# In Vitro and In Vivo Chaperone Effect of (R)-2-amino-6-(1R, 2S)-1,2-dihydroxypropyl)-5,6,7,8-tetrahydropterin-4(3H)-one on the C1473G Mutant Tryptophan Hydroxylase 2

**DOI:** 10.3390/biom13101458

**Published:** 2023-09-27

**Authors:** Alla B. Arefieva, Polina D. Komleva, Vladimir S. Naumenko, Nikita V. Khotskin, Alexander V. Kulikov

**Affiliations:** 1Department of Genetic Collections of Neural Disorders, Federal Research Center Institute of Cytology and Genetic Siberian Branch of Russian Academy of Sciences, 630090 Novosibirsk, Russia; a.arefeva@alumni.nsu.ru (A.B.A.); khotskin@bionet.nsc.ru (N.V.K.); 2Department of Psychoneuropharmacology, Federal Research Center Institute of Cytology and Genetic Siberian Branch of Russian Academy of Sciences, 630090 Novosibirsk, Russia; komleva@bionet.nsc.ru (P.D.K.); naumenko2002@bionet.nsc.ru (V.S.N.); 3Departments of Behavioral Neurogenomics, Federal Research Center Institute of Cytology and Genetic Siberian Branch of Russian Academy of Sciences, 630090 Novosibirsk, Russia

**Keywords:** serotonin, tryptophan hydroxylase 2, tetrahydrobiopterin, thermal stability, behavior, brain, mice

## Abstract

Tryptophan hydroxylase 2 (TPH2) is the key and rate-limiting enzyme of serotonin (5-HT) synthesis in the mammalian brain. The 1473G mutation in the Tph2 gene decreases TPH2 activity in the mouse brain by twofold. (R)-2-amino-6-(1R, 2S)-1,2-dihydroxypropyl)-5,6,7,8-tetrahydropterin-4(3H)-one (BH_4_) is a pharmacological chaperone for aromatic amino acid hydroxylases. In the present study, chaperone effects of BH_4_ on the mutant C1473G TPH2 were investigated in vitro and in vivo. In vitro BH_4_ increased the thermal stability (T_50_ value) of mutant and wild-type TPH2 molecules. At the same time, neither chronic (twice per day for 7 days) intraperitoneal injection of 48.3 mg/kg of BH_4_ nor a single intraventricular administration of 60 μg of the drug altered the mutant TPH2 activity in the brain of Balb/c mice. This result indicates that although BH_4_ shows a chaperone effect in vitro, it is unable to increase the activity of mutant TPH2 in vivo.

## 1. Introduction

Tryptophan hydroxylase 2 (TPH2) hydroxylates L-tryptophan to L-5-hydroxytryptophan in the first rate-limiting stage of serotonin (5-HT) synthesis in the mammalian brain [1,2]. The *Tph2* gene knockout [3,4,5] or the TPH2 inhibitor, p-chlorophenylalanine [6,7], dramatically reduces the 5-HT level in the mouse brain. Mutations in the human *Tph2* gene are associated with risk of depressive disorders [8,9].

TPH2 is a member of the aromatic amino acid hydroxylase family that, besides TPH2, includes three other enzymes: tryptophan hydroxylase 1 (TPH1), tyrosine hydroxylase (TH) and phenylalanine hydroxylase (PAH) [10]. These three enzymes hydroxylate L-tryptophan, L-tyrosine and L-phenylalanine, respectively, in the presence of reduced tetrahydrobiopterin (BH_4_) cofactor and O_2_ [10]. Some single-nucleotide mutations in the *Th* and *Pah* genes cause nonsynonymous single-amino-acid substitutions that lead to misfolding of 3D structures and decreases in TH and PAH activities [11,12,13].

Pharmacological chaperones are structure-directed small molecules that specifically stabilize the native state of the target protein and/or facilitate the folding of non-native intermediate states toward the native protein [11,12,13]. BH_4_ shows a chaperone effect and stabilizes some mutant TH and PAH molecules in vitro [11,12,13]. Chronic treatment with BH_4_ increases TH activity in the mouse brain [14].

Several mutations in the *Tph2* gene suggest a decrease in TPH2 activity via truncation of the enzyme molecule [15], as well as a decrease its expression [16,17] or stability [18,19,20]. It can be hypothesized that negative effects of at least the last mutations altering the folding and stability of the TPH2 molecule could be corrected through BH_4_ therapy.

Single functional amino acid substitutions in mouse TPH2 molecules decreasing the enzyme activity seem to be promising instruments for modeling of the in vitro and in vivo effects of BH_4_ treatment on the enzyme, as well as behavioral disorders caused by *Tph2* gene mutations. There are two mutations in the mouse *Tph2* gene that dramatically decrease the enzyme activity: the G1449A and C1473G polymorphisms result in R439H and P447R substitution in the TPH2 molecule, respectively. The first was generated as the mouse homolog of the G1463A polymorphism in the human Tph2 gene [21]. The second is the spontaneous mutation found in laboratory mice [22,23,24]. Mice homozygous for the 1473G allele exhibit a twofold decrease in TPH2 activity in their brain compared to those homozygous for the 1473C allele [22,23,24]. Only the 1473C allele is found in natural populations of wild mice; therefore, it represents the wild-type allele, while the 1473G is a mutant allele [25].

The C1473G polymorphism seems to be a convenient model for studying the chaperone effect of BH_4_ on mutant TPH2 molecules. First, this polymorphism is the main genetic factor defining the hereditary variability in TPH2 activity in the brain of laboratory mice [26]. Second, in our preliminary study, we showed that the mutant 1473G allele decreased the enzyme stability. Third, this allele causes hind-leg dystonia in mice [27]. Finally, both alleles are available for research, since they are found in common strains of laboratory mice: the wild-type 1473C allele is found in C57BL/6, CBA, AKR and C3H strains, while the mutant 1473G allele is found in Balb/c and DBA2 strains [23].

The aim of the present study was to test the chaperone effect of BH_4_ on the mutant C1473G TPH2 in vitro and in vivo. We plan to answer the following questions: (1) whether BH_4_ can increase the thermal stability of wild-type and mutant TPH2 molecules in vitro, (2) whether BH_4_ can penetrate the brain after an intraperitoneal administration, (3) whether chronic intraperitoneal administration of BH_4_ can restore decreased TPH2 activity and hind-limb dystonia caused by the 1473G allele and (4) whether a single intraventricular BH_4_ administration can affect TPH2 activity and 5-HT metabolism in the mouse brain.

## 2. Materials and Methods

### 2.1. Animals

This study was conducted in strict accordance with the recommendations of Directive 2010/63/EU of the European Parliament and of the Council of 22 September 2010 on the protection of animals used for scientific purposes and was approved by the Committee on the Ethics of Animal Experiments of the Russian National Center of Genetic Resources of Laboratory Animals of Institute of Cytology and Genetics, Siberian Branch of Russian Academy of Sciences (Approval Code: Protocol N 139. Approval Date: 7 December 2022). Mice were bred in the Collective Centre of Animal Genetic Resources (supported by basic research project Nos. FWNR-2022-0023 and RFMEFI62117X0015).

Experiments were carried out on SPF-state 12-week-old male CD-1 (CC), Balb/c (GG) and C57BL/6 (CC) mice. Balb/c and C57BL/6 mice were chosen as models of mutant and wild-type genotypes, respectively. CD-1 mice were chosen as a “neutral” model to study the ability of BH_4_ to penetrate the brain. During all experiments, the animals were kept separately in cages (Optimice, Animal Care Systems, Inc., Centennial, CO, USA) at a temperature of 24 ± 2 °C and humidity of 45–50%, with an artificial 14:10 (light: dark) photoperiod with daybreak and sunset at 01:00 and 15:00, respectively. The mice were fed sterile food and water ad libitum.

### 2.2. Drug and Treatment

For the in vitro experiment, BH_4_ ((R)-2-amino-6-(1R, 2S)-1,2-dihydroxypropyl)-5,6,7,8-tetrahydropterin-4(3H)-one 2 HCl, abcr GmbH, Karlsruhe, Germany) was dissolved in sterile ultrapure water at a final concentration of 30 mM. For the intraperitoneal injection, BH_4_ was dissolved in a final concentration of 48.3, 96.6 or 194.2 mg/10 mL in sterile saline, corresponding to doses of 48.3 (0.2 mM), 96.6 (0.4 mM) or 194.2 (0.8 mM) mg/kg of body mass, respectively. For the intraventricular injection, BH_4_ was dissolved in a final concentration of 1 mg/50 μL of sterile saline. All solutions were stored at −20 °C until use.

### 2.3. Experimental Design and Treatment

The following five experiments were performed.

In experiment 1, the expression of hind-limb dystonia in a tail suspension test was compared in adult male Balb/c (n = 8) and C57BL/6 (n = 8) mice.

In experiment 2, the effect of BH_4_ on in vitro thermal stability of wild-type and mutant TPH2 molecules extracted from the midbrain of C57BL/6 (n = 6) and Balb/c (n = 6) male mice, respectively, was studied.

In experiment 3, the ability of BH_4_ to penetrate the brain after its intraperitoneal administration was investigated. Four experimental groups of five CD-1 males were formed: injected with (1) saline, (2) 48.3, (3) 96.6 or (4) 194.2 mg/kg of BH_4_. One hour after the injection, the mice were euthanized by carbon dioxide asphyxiation followed by decapitation. Their hippocampus and midbrain were rapidly dissected, frozen in liquid nitrogen and stored at −80 °C until the BH_4_ level assay in these structures.

In experiment 4, the effects of chronic intraperitoneal administration of BH_4_ on hind-limb dystonia, 5-HT, 5-HIAA levels and TPH2 activity in Balb/c mice were studied. Two experimental groups of eight Balb/c males were formed: treated with (1) saline and (2) 48.3 mg/kg of BH_4_. They were injected twice per day (at 10:00 and 15:00) for seven successive days (or 14 injections) with saline (group 1) or BH_4_ (group 2). All animals were tested for hind-limb dystonia before the 1st injection and after the 12th one. On the 8th day of the experiment, the mice were euthanized by carbon dioxide asphyxiation followed by decapitation. Their hippocampus and midbrain were rapidly dissected, frozen in liquid nitrogen and stored at −80 °C until the assay of 5-HT, 5-HIAA levels and TPH2 activity in these structures.

In experiment 5, the effects of a single intraventricular administration of BH_4_ on 5-HT, 5-HIAA levels and TPH2 activity in Balb/c and C57BL/6 mice were analyzed. Four groups of eight mice were formed: (1) C57BL/6, saline; (2) C57BL/6, BH_4_; (3) Balb/c, saline; (4) Balb/c, BH_4_. Then, 3 μL of saline or BH_4_ solution (60 μg) was injected into a lateral ventricle of the mice with a Hamilton syringe under light isoflurane anesthesia, and 24 h after the injection, the mice were euthanized by carbon dioxide asphyxiation followed by decapitation. Their hippocampus and midbrain were rapidly dissected, frozen in liquid nitrogen and stored at −80 °C until the assay of 5-HT, 5-HIAA levels and TPH2 activity in these structures.

### 2.4. Tail Suspension Test

In the present study, a tail suspension test was used to measure the number and accumulated time of hind-limb clasping [28]. Mice were fixed by their tails using adhesive tape and hooked to a horizontal bar placed 30 cm above the table surface. The number and accumulated time of hind-limb clasping was registered by an experienced rater [27].

### 2.5. Tissue Preparation

In experiment 2, TPH2 was extracted from the midbrain of C57BL/6 (n = 6) and Balb/c (n = 6) mice. The midbrain was homogenized in 400 uL of cold 50 mM Tris HCl (pH 7.6) and 1 mM dithiotreitol using a motor-driven grinder (Z359971, Sigma-Aldrich, Darmstadt, Germany) and spun for 15 min at 12,700 rpm (+4 °C). The clear supernatant from six midbrains of mice of the same genotypes was pulled, aliquoted in 300 uL portions and stored at −80 °C until the TPH2 activity assay (see Section 2.7).

In experiments 4 and 5 brain samples used to assay BH_4_, serotonin (5-HT), 5-hydroxyindoleacetic acid (5-HIAA) levels and TPH2 activity in the midbrain and hippocampus as structures containing 5-HT neuron bodies and endings, respectively. TPH2 molecules are synthesized in 5-HT neuron bodies (in midbrain), then transported to 5-HT neuron endings [24]. The hippocampus and midbrain of the mice were homogenized in 400 μL of 50 mM Tris HCl (pH 7.6) and 1 mM dithiotreitol using a motor-driven grinder (Z359971, Sigma-Aldrich, Darmstadt, Germany). One aliquot of 50 μL of the homogenate was mixed with 150 μL of 0.6 M HClO_4_ for the extraction of BH_4_, 5-HT and 5-HIAA (see Section 2.6). The rest of homogenate was spun for 15 min at 12,700 rpm (+4 °C), and the clear supernatant was stored at −80 °C until the TPH2 activity assay (see Section 2.7).

### 2.6. Assay of BH_4_, 5-HT and 5-HIAA Levels

The mix of 50 μL of homogenate with 150 μL of 0.6 M HClO_4_ (see Section 2.5) was spun for 15 min at 12,700 rpm (+4 °C). The pellet was dissolved in 1 mL of 0.1 M NaOH and used for protein determination using the Bradford method (Bio Rad, Hercule, CA, USA). The clear supernatant was diluted twofold with ultrapure water, and BH_4_, 5-HT and 5-HIAA levels were assayed in the diluted supernatant using the HPLC technique on a Luna C18(2) column (5 μm particle size, L × I.D. 100 × 4.6 mm, Phenomenex, Torrance, CA, USA) with electrochemical detection (750 mV, DECADE II™ Electrochemical Detector; Antec, Alphen aan den Rijn, The Netherlands), a glassy carbon flow cell (VT-03 cell 3mm GC sb; Antec, The Netherlands), CBM-20 A system controller, LC-20AD solvent delivery unit, SIL-20 A autosampler and DGU-20A5R degasser (Shimadzu Corporation, Kyoto, Japan). A volume of 1 L of the mobile phase (pH = 3.2) contained 13.06 g of KH_2_PO_4_, 200 μL of 0.5 M Na_2_EDTA, 300 mg of 1-octanesulfonic acid sodium salt (Sigma-Aldrich, Darmstadt, Germany), 940 μL of concentrated H_3_PO_4_ and 130 mL of methanol (13% volume; Vektor Ltd., Saint-Petersburg, Russia) [27].

Standard mixes containing 1, 2 and 3 ng of BH_4_, 5-HT and 5-HIAA, respectively, were repeatedly assayed throughout the entire procedure and used to plot the calibration curves for each substance. The areas of peaks were estimated using LabSolution LG/GC software version 5.54 (Shimadzu Corporation, Kyoto, Japan) and calibrated against the calibrated curves for corresponding standards. BH_4_, 5-HT and 5-HIAA contents were expressed in ng/mg of protein [27]. All data are presented as means of three replications.

### 2.7. Assay of TPH Activity

An aliquot of 15 μL of pure supernatant (see Section 2.5) was incubated for 15 min at 37 °C in the presence of L-tryptophan (Sigma-Aldrich, Darmstadt, Germany) (0.4 mM), cofactor 6-methyl-5,6,7,8-tetrahydropteridine (Sigma-Aldrich, Darmstadt, Germany) (0.3 mM), decarboxylase inhibitor m-hydroxybenzylhydrazine (Sigma-Aldrich, Darmstadt, Germany) (0.3 mM), catalase (Sigma-Aldrich, Darmstadt, Germany) (5 U) and 1 mM dithiothreitol in a final volume of 25 μL. The reaction was stopped with 75 μL 0.6 M HClO_4_ and centrifuged for 15 min at 12,700 rpm. The clear supernatant was diluted twofold with ultrapure water. The amount of synthesized 5-hydroxytryptophan (5-HTP) was determined in the diluted supernatant using HPLC (see Section 2.6) and calibrated against the calibrated curves for corresponding standards of 25, 50 and 100 pmoles of 5-HTP (Sigma-Aldrich, Darmstadt, Germany). Another aliquot of 10 μL of supernatant was mixed with 90 μL of 0.1 M NaOH for protein quantitation using the Bradford method (Bio Rad, Hercule, CA, USA) according to the manufacturer’s protocol. The TPH activity was measured as pmoles of 5-HTP formed per minute per mg of protein [27].

### 2.8. Assay of BH_4_ Effect on TPH2 Thermal Stability

First, 10 μL aliquots of pure supernatant (see Section 2.5) were mixed with 5 μL of 50 mM Tris HCl buffer (pH 7.6) containing 1 mM of dithiothreitol or with 0.6 mM BH_4_ solution in this buffer (the final concentration of BH_4_ was 0.2 mM) and heated for 2 min at 48, 50, 52, 54. 56, 58 and 60 °C. The control tubes were not heated. Then, 10 μL of the mix of L-tryptophan (1 mM), 6-methyl-5,6,7,8-tetrahydropteridine (0.75 mM), m-hydroxybenzylhydrazine (0.75 mM), catalase (5 U), 1 mM dithiothreitol (the final concentrations of these chemicals were 0.4, 0.3 and 0.3 mM, respectively) was added, and the amount of synthesized 5-HTP was assayed after a 15 min incubation at 37 °C (see Section 2.7). Four groups of 4–5 thermal curves each were formed: (1) C57BL/6 without BH_4_, (2) C57BL/6 with BH_4_, (3) Balb/c without BH_4_ and (4) Balb/c with BH_4_. The linear parts of these curves were used to calculate T_50_ values using a linear regression method.

### 2.9. Statistics

The number and duration of hind-limb claspings in experiment 1; T_50_ values (experiment 2); BH_4_, 5-HT and 5-HIAA levels; and TPH2 activity (experiments 4 and 5) were presented as means ± SEM and analyzed using one-way (experiment 4) or two-way ANOVA (experiments 1 and 5) with “genotype” and “treatment” as independent factors, as well as their interaction. The number and duration of hind-limb claspings in experiment 4 are presented as means ± SEM and were analyzed using two-way ANOVA for repeated measures with “treatment” as a between factor and “duration” as a within factor, as well as their interaction. Post hoc analyses were carried out using the Fisher’s LSD multiple comparison test when appropriate. The BH_4_ concentration in the brain in experiments 4 and 5 was compared with zero using Student’s t-test. Statistical significance was set at *p* < 0.05.

The relationship between the injected amount and the final level of BH_4_ in the brain (experiment 3) was analyzed by applying a linear regression: Y = b × X + a, where “Y” is the level of BH_4_ in the brain (ng/mg of tissue), “X” is the amount of drug injected intraperitoneally (mg/kg of body mass), and “b” and “a” are the regression coefficients. The software calculates both “b” and “a” mean values and their errors.

## 3. Results

### 3.1. Hind-Limb Dystonia in Balb/c and C57BL/6 Mice (Experiment 1)

A marked increase in the number (C57BL/6, 8.6 ± 2.6; Balb/c, 26.1 ± 3.5; F(1,14) = 15.7, *p* = 0.0014) and accumulated time (C57BL/6, 31.4 ± 10.6 s; Balb/c, 211.2 ± 22.8 s; F(1,14) = 51.2, *p* < 0.001) of hind-limb claspings in Balb/c mice compared to C57BL/6 mice was shown.

### 3.2. Effect of BH_4_ on Thermal Stability (T_50_) of TPH2 Extracted from Midbrain of C57BL/6 and Balb/c Mice (Experiment 2)

Marked effects of “genotype” (F(1,16) = 373.1, *p* < 0.001) and “treatment” (F(1,16) = 100.7, *p* < 0.001) factors—but not their interaction (F(1,16) = 1.6, *p* = 0.23)—on T_50_ value were shown for TPH2 extracted from the midbrain of C57BL/6 and Balb/c mice. As expected, the T_50_ value for TPH2 from C57BL/6 mice was higher than that of Balb/c mice (*p* < 0.001, Figure 1). BH_4_ increased the T_50_ value for TPH2 extracted from C57BL/6 (*p* < 0.001) and Balb/c (*p* < 0.001) mice (Figure 1).

### 3.3. BH_4_ Level in Hippocampus and Midbrain of CD-1 Mice an Hour after a Single Intraperitoneal Administration of This Drug (Experiment 3)

A single intraperitoneal administration of BH_4_ markedly increased the drug level in the hippocampus (F(3,16) = 180.0, *p* < 0.001) and the midbrain (F(3,16) = 39.3, *p* < 0.001) in CD-1 mice. The BH_4_ levels in the hippocampus (BH4 level = BH4 administered × 0.0032 + 0.29) and the midbrain (BH4 level = BH4 administered × 0.0035 + 0.54) increased linearly with the administered dose (Figure 2). The regression coefficients for the hippocampus (0.0032 ± 0.0002) and the midbrain (0.0035 ± 0.0003) indicate that only 0.32% and 0.35% of the intraperitoneally administered BH_4_ penetrated these structures.

### 3.4. Effect of Chronic BH_4_ Intraperitoneal Administration on Hind-Limb Dystonia; BH_4_, 5-HT and 5-HIAA Levels; and TPH2 Activity in the Hippocampus and Midbrain of Balb/c Mice (Experiment 4)

No effects of the “treatment” and “duration” factors and their interaction on the number and accumulating time of hind-limb claspings were observed in Balb/c mice in the tail suspension test (Table 1). Chronic intraperitoneal administration of 48.3 mg/kg of BH_4_ two times per day for 6 days failed to affect their number and accumulated time of hind-limb clasping (Figure 3).

Fourteen intraperitoneal injections for 7 successive days with 48.3 mg/kg of BH_4_ did not significantly increase the drug level in the hippocampus (0.061 ± 0.115 ng/mg, t(7) = 0.53, *p* = 0.61) or the midbrain (0.502 ± 0.370 ng/mg, t(7) = 1.36, *p* = 0.22).

No difference in 5-HT and 5-HIAA levels in the hippocampus (5-HT, F(1,14) = 3.32, *p* = 0.09; 5-HIAA, F(1,14) < 1) and the midbrain (5-HT, F(1,14) < 1; 5-HIAA, F(1,14) < 1) between Balb/c mice treated twice per day for 7 days with saline and 48.3 mg/kg of BH_4_ was observed (Figure 4).

No difference in the TPH2 activity in the hippocampus (F(1,14) = 1.15, *p* = 0.30) and the midbrain (F(1,14) = 2.67, *p* = 0.12) between Balb/c mice treated ip twice per day for 7 days with saline and 48.3 mg/kg of BH_4_ was observed (Figure 5).

### 3.5. Effect of Acute BH_4_ Intraventricular Administration on BH_4_, 5-HT and 5-HIAA Levels and TPH2 Activity in yjr Hippocampus and Midbrain of Balb/c and C57BL/6 Mice (Experiment 5)

Significant increases in BH_4_ levels in the hippocampus (0.57 ± 0.22 ng/mg, t(7) = 2.55, *p* = 0.038) and the midbrain (0.69 ± 0.19 ng/mg, t(7) = 3.75, *p* = 0.007) in C57BL/6 mice 24 h after a single intraventricular administration of 60 μg of the drug were observed. At the same time, in Balb/c mice, 24 h after a single intraventricular administration of 60 μg of BH_4_, a significant increase in the drug level was observed only in the midbrain (0.74 ± 0.30 ng/mg, t(7) = 2.44, *p* = 0.045) but not in the hippocampus (0.49 ± 0.23 ng/mg, t(7) = 2.10, *p* = 0.073).

A significant effect of the “genotype” and “treatment” factors but not their interaction on the 5-HT level in the hippocampus and the midbrain was revealed. At the same time, only a significant effect of the “genotype” factor on 5-HIAA in the hippocampus and the midbrain of mice was shown (Table 2). 5-HT (*p* < 0.001) and 5-HIAA (*p* < 0.001) levels were higher in the hippocampus of the saline-treated C57BL/6 mice compared to the saline-treated Balb/c mice (Figure 6). Intraventricular administration of BH_4_ decreased the 5-HT level in the midbrain of Balb/c mice and increased the 5-HIAA level in this structure in C57BL/6 mice (Figure 6).

A significant effect of the “genotype” factor (hippocampus, F(1,27) = 27.8, *p* < 0.001; midbrain F(1,27) = 74.3, *p* < 0.001) but not the “treatment” factor (hippocampus, F(1,27) = 1.03, *p* = 0.32; midbrain F(1,27) < 1) factor or their interaction (hippocampus, F(1,27) = 1.16, *p* = 0.29; midbrain F(1,27) = 1.89, *p* = 0.18) on the TPH2 activity in the hippocampus and the midbrain was shown. As expected, the TPH2 activity in these structures in the saline-treated C57BL/6 mice was higher than that in the saline-treated Balb/c mice (Figure 7). No effect of BH_4_ administration on the TPH2 activity in these structures was observed in either strain (Figure 7).

## 4. Discussion

BH_4_ is the natural cofactor and H^+^ donor for key metabolic enzymes such as all hydroxylases, aromatic amino acids, TPH1, TPH2, TH, PAH, all isoforms of NO synthases (NOS I, II and III) and alkylglycerol monooxygenase [29]. Its deficiency increases the risk of some grave mental diseases, such as phenylketonuria, autism spectrum diseases, and Parkinson’s and Alzheimer’s diseases [29,30]. A commercial version of BH_4_ called “sapropterin” is used for treatment of some disorders [29,30,31,32,33,34,35].

In addition to its function as a cofactor, BH_4_ shows a chaperone effect and corrects the misfolding of mutant TH and PAH molecules [11,12,13]. It increases the thermal stability of mutant TH and PAH molecules in vitro [11,12,13]. Moreover, chronic (for 5 weeks) peroral administration of 100 mg/kg/day of BH_4_ increases the TH activity in the brain of C57BL/6 mice [14]. However, a chaperone-like effect of BH_4_ on mutant TPH2 molecules remains obscure. The C1473G polymorphism in the murine *Tph2* gene resulting in P447R substitution in the TPH2 molecule catalytic domain and about twofold reduction in the enzyme activity in the mouse brain [22,23,24] is a convenient model to search for a possible chaperone-like effect of BH_4_ on the TPH2 molecule.

The main aim of the present study was to investigate the ability of BH_4_ to correct the mutant TPH2 activity in vitro and in vivo. Mice of the Balb/c line homozygous for 1473G allele were chosen as a model object to study the chaperone effect of BH_4_ on the mutant TPH2 molecule, while C57BL/6 line mice homozygous for the 1473C allele were chosen as a model of the wild-type enzyme molecule.

Some single-amino-acid substitutions cause misfolding and decrease the stability of affected (mutant) molecules, while chaperones correct folding and restore the stability of the mutant molecules. The stability of molecules is assayed in vitro according to their thermal denaturation, which is usually evaluated by the fluorescence of the complex of target molecules with 8-anilino-1-nathalener sulfonic acid. Half-denaturation temperature (T_50_) is the T value at which half of the molecules are in the unfolded state [11,12,13]. Since this technique requires pure recombinant protein and it is not applicable for a mix of different proteins, we developed an alternative technique for a thermal denaturation assay by monitoring the decrease in the TPH2 activity resulting from preheating of brain extracts at 48–60 °C before an enzyme activity assay at 37 °C. This technique does not need pure recombinant enzyme molecules and is applicable to brain extracts.

Here, a decrease in thermal stability (decrease in T_50_ value) of mutant TPH2 molecules (from Balb/c) compared to the wild-type molecule (from C57BL/6) was found, indicating that the 1473G allele causes misfolding of the mutant TPH2 molecule. BH_4_ acts as pharmacological chaperone and increases the T_50_ value for both mutant and wild-type TPH2 molecules in vitro.

The next aim was to test the ability of BH_4_ administered peripherally (intraperitoneally) to penetrate the brain. For this purpose, CD-1 mice were used as a “neutral” model object. The BH_4_ level in the brain increases linearly with the drug dose injected intraperitoneally. However, the drug rate found in the brain is about 0.3% of its intraperitoneally administered amount. Therefore, BH_4_ administered intraperitoneally hardly penetrates the brain.

A logical question arises: is the small amount of BH_4_ that still penetrates the brain enough to increase the TPH2 activity and reduce the expression of dystonia caused by the mutation? Chronic intraperitoneal administration of 48.3 mg/kg twice per day (or 96.6 mg/kg/day) for 6 and 7 successive days failed to decrease hind-limb dystonia and to increase the TPH2 activity in the brain of Balb/c mice. This treatment also did not alter 5-HT or 5-HIAA levels. Moreover, no detectable amount of BH_4_ was found in these brain structures.

The reported results may suggest that the treatment duration employed in our experiment (one week) is not sufficient to correct the mutant TPH2 molecules in the mouse brain. However, the half-life of TPH2 in the rat brain is about 48 h [36,37]; therefore, a weeklong period is sufficient to update the enzyme molecules in the brain. Therefore, our negative result indicates that intraperitoneal BH_4_ administration cannot increase TPH2 activity, 5-HT levels and metabolism or correct the behavioral disorder caused by low TPH2 activity due to an extremely low level of drug penetration in the brain.

The last question that we attempted to answer in this study was whether BH_4_ injected directly into the brain (intraventricular administration) can restore the deficit of the TPH2 activity caused by the C1473G mutation. The concentration of BH_4_ observed in the brain 24 h after a single intraventricular administration of 60 μg of BH_4_ was about 0.5–0.7 ng/mg of protein. Since the total protein concentration is about 10% of the brain mass and the adult mouse brain mass is about 400–500 mg [38], the BH_4_ amount in the mouse brain 24 h after the administration of 60 μg of BH_4_ is only 2–3 μg or 3–5% of the injected amount, corresponding to about 0.02–0.03 mM. These results agree with the half-life of BH_4_ in tissue, which is about 4 h [39]. This in vivo drug concentration is too low compared to its effective chaperone concentration in vitro (0.2 mM). It was no surprise that we did not observe any alteration in the TPH2 activity in the hippocampus or the midbrain in Balb/c and C57BL/6 mice 24 h after a single intraventricular administration of 60 μg of BH_4_. This administration even decreased 5-HT levels in these structures.

Thus, in vitro BH_4_ shows its chaperone activity and increases the thermal stability of the mutant TPH2 molecule. At the same time, this drug does not seem to be effective in the treatment of brain disorders in vivo. First, when injected intraperitoneally, BH_4_ hardly penetrates the brain. Secondly, its lifespan in the brain tissue is too short. These two obstacles prevent a high concentration of the drug from remaining in the brain for a long time that would be sufficient for its chaperone effect.

However, the problem of therapeutic efficacy of exogenous BH_4_ seems to be more complex. Some authors showed an increase in 5-HT and dopamine levels in the brain after chronic intraperitoneal and subcutaneous administration of the drug (see review [40]). Moreover, chronic (5 weeks) peroral administration of 100 mg/kg of BH_4_ increased TH activity in the mouse brain [14] and improved recognition memory in a mouse model of Alzheimer’s disease [41]. These results obtained by several independent research groups cannot be ignored, and this problem requires additional studies of the pharmacokinetics of exogenously administered BH_4_.

Two alternative solutions to this problem can be proposed: (1) treatment with an artificial chemical analogous to BH_4_ that shows chaperone activity in vitro, a long lifespan in tissue and the ability to penetrate the brain in vivo; (2) an increase in the endogenous BH_4_ level in the brain, for example, by administration of BH_4_ metabolic precursors.

## 5. Conclusions

The present work is a logical development of previous studies concerning the effects of the C1473G mutation in the *Tph2* gene on the mouse brain and behavior [26], as well as in vitro and in vivo chaperone effects of BH_4_ [11,12,13]. Here, the effect of this mutation on hind limbs dystonia, as well as chaperone effects of BH_4_ on in vitro thermal stability and in vivo activity of mutant TPH2 molecules, was investigated. The following results were obtained:In the tail suspension test, the mutant 1473G allele increased the frequency and duration of hind-limb dystonia in Balb/c mice compared to the wild-type 1473C allele in C57BL/6 mice.In vitro, the 1473G allele decreased the thermal stability of the TPH2 molecule (decreased T_50_ value) compared to the 1473C allele. BH_4_ at a concentration of 0.2 mM increased the thermal stability (increases T_50_ value) of mutant and wild-type TPH2 molecules.Only 0.32–0.35% of BH_4_ penetrated the murine brain an hour after intraperitoneal drug administration.Chronic intraperitoneal administration of 48.3 mg/kg of BH_4_ twice per day for 7 days failed to decrease hind-leg dystonia and alter 5-HT and 5-HIAA levels and TPH2 activity in the hippocampus and midbrain of Balb/c mice (homozygous for the 1473G allele).A single intraventricular administration of 60 μg of BH_4_ failed to alter the TPH2 activity in the hippocampus and the midbrain in Balb/c and C57BL/6 mice.

These results indicate that although BH_4_ shows a chaperone-like effect and increases thermal stability of mutant TPH2 molecules in vitro, it is unable to decrease hind-leg dystonia and increase TPH2 activity in vivo in Balb/c mice homozygous for the 1473G allele.

## Figures and Tables

**Figure 1 biomolecules-13-01458-f001:**
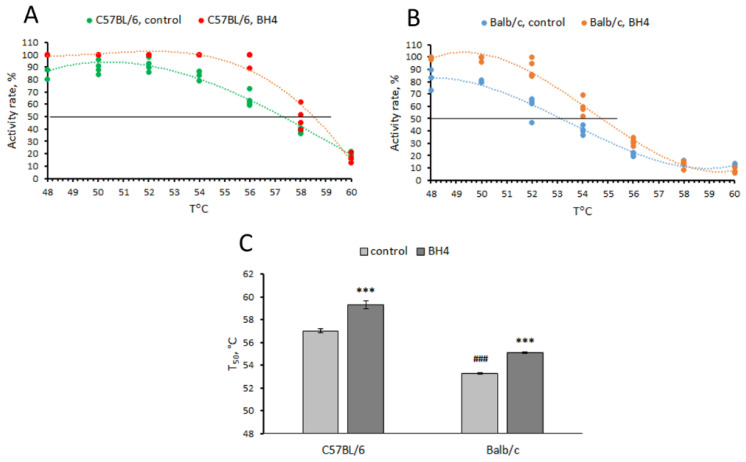
Thermal denaturation curves (**A**,**B**) and T_50_ values (**C**) for TPH2 extracted from the midbrain of C57BL/6 (CC) and Balb/c (GG) males assayed in the absence (control) or presence (BH_4_) of 0.2 mM of BH_4_. *** *p* < 0.001 vs. corresponding control; ### *p* < 0.001 vs. C57BL/6 control.

**Figure 2 biomolecules-13-01458-f002:**
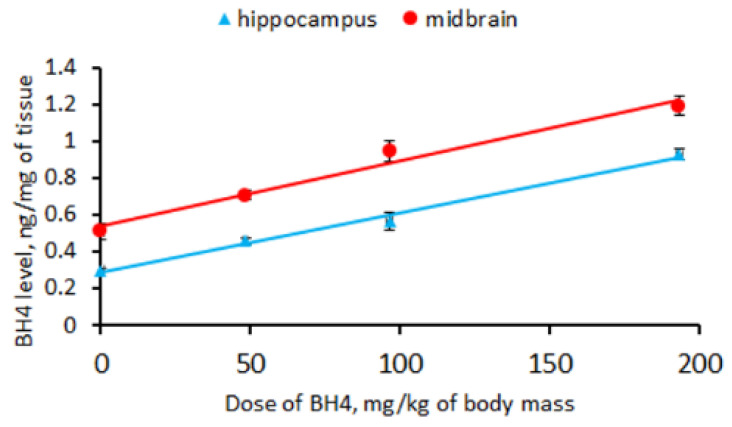
Linear regression between the BH_4_ level (ng/mg) in the hippocampus and midbrain and the drug dose (mg/kg) administered intraperitoneally to CD-1 mice.

**Figure 3 biomolecules-13-01458-f003:**
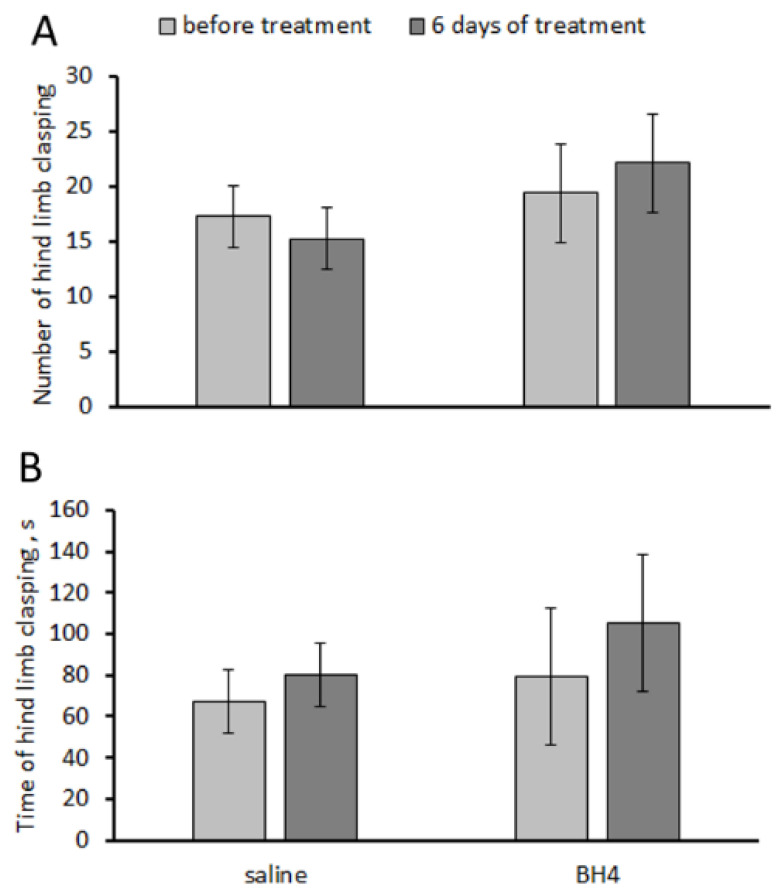
Number (**A**) and accumulated time (s) (**B**) of hind-limb claspings in the tail suspension test in Balb/c mice before and after chronic intraperitoneal treatment with saline and 48.3 mg/kg of BH_4_ twice per day for 6 days.

**Figure 4 biomolecules-13-01458-f004:**
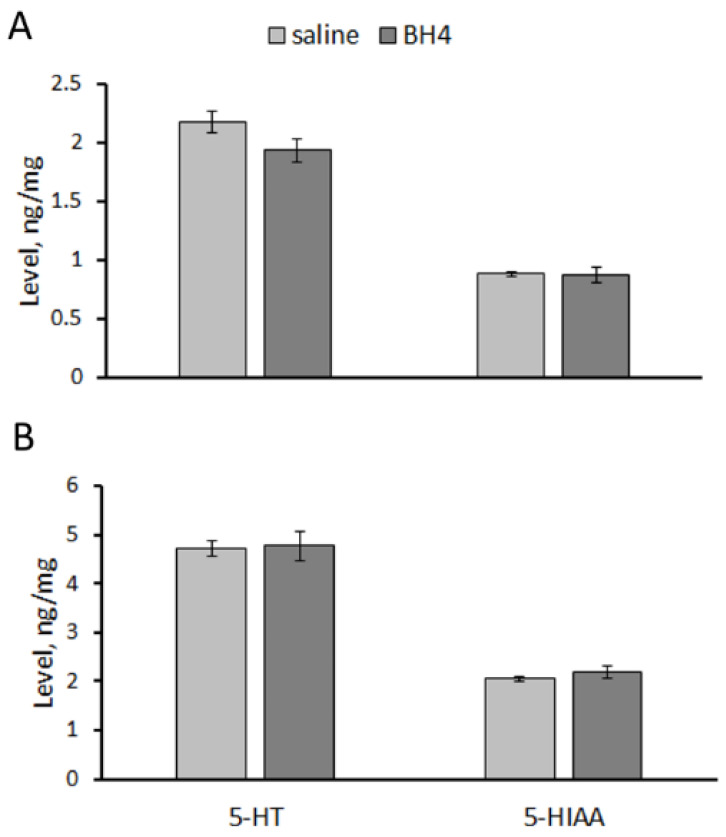
5-HT and 5-HIAA levels (ng/mg) in the hippocampus (**A**) and midbrain (**B**) of Balb/c mice treated ip twice per day for 7 successive days with saline or 48.3 mg/kg of BH_4_.

**Figure 5 biomolecules-13-01458-f005:**
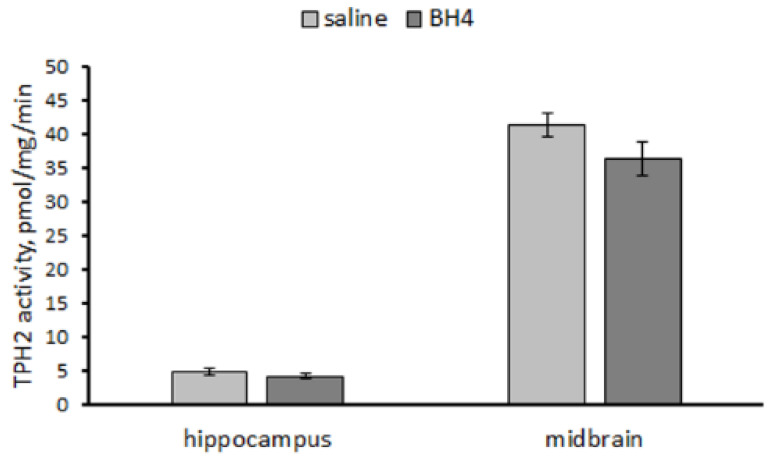
TPH2 activity (pmol/mg/min) in the hippocampus and midbrain in Balb/c mice treated ip twice per day for 7 successive days with saline or 48.3 mg/kg of BH_4_.

**Figure 6 biomolecules-13-01458-f006:**
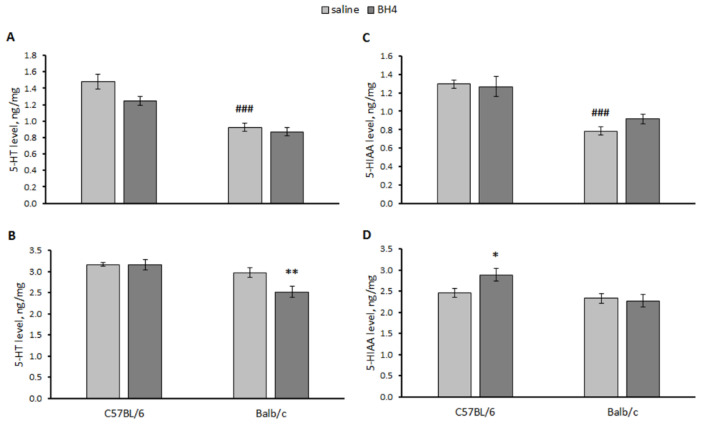
5-HT and 5-HIAA levels (ng/mg) in the hippocampus (**A**,**C**) and midbrain (**B**,**D**) of C57BL/6 and Balb/c mice 24 h after a single intraventricular administration of saline or 60 μg of BH_4_. ### *p* < 0.001 vs. saline-treated C57BL/6 mice. * *p* < 0.05, ** *p* < 0.01 vs. corresponding saline-treated control.

**Figure 7 biomolecules-13-01458-f007:**
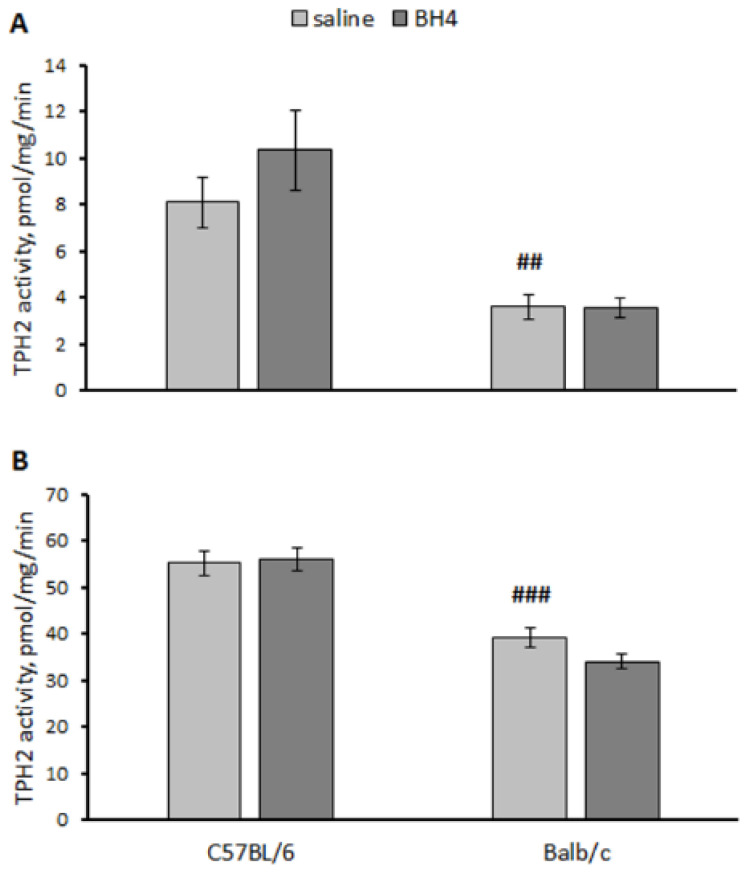
TPH2 activity in the hippocampus (**A**) and midbrain (**B**) in C57BL/6 and Balb/c mice 24 h after intraventricular administration of saline or 60 μg of BH_4_. ## *p* < 0.01, ### *p* < 0.001 vs. saline-treated C57BL/6 mice.

**Table 1 biomolecules-13-01458-t001:** Two-way ANOVA of the effects of “treatment” and “duration” factors and their interaction on the variability of the number and accumulated time in the tail suspension test in Balb/c mice treated ip twice per day for 6 days with saline or 48.3 mg/kg of BH_4_.

Factor	Number	Accumulated Time
“Treatment”	F(1,14) = 1.2, *p* = 0.30	F(1,14) < 1
“Duration”	F(1,14) < 1	F(1,14) < 1
“Treatment” × “Duration”	F(1,14) < 1	F(1,14) < 1

**Table 2 biomolecules-13-01458-t002:** Two-way ANOVA of “genotype” and “treatment” factors and their interaction on the variability of 5-HT and 5-HIAA levels in the hippocampus and midbrain in C57BL/6 and Balb/c mice 24 h after a single intraventricular administration of 60 μg of BH_4_.

Factor	5-HT	5-HIAA
Hippocampus		
“Genotype”	F(1,27) = 60.4, *p* < 0.001	F(1,27) = 37.2, *p* < 0.001
“Treatment”	F(1,27) = 5.67, *p* = 0.025	F(1,27) < 1
“Genotype” × “Treatment”	F(1,27) = 2.16, *p* = 0.15	F(1,27) = 1.22, *p* = 0.22
Midbrain		
“Genotype”	F(1,27) = 13.34, *p* = 0.001	F(1,27) = 7.38, *p* = 0.011
“Treatment”	F(1,27) = 4.21, *p* = 0.049	F(1,27) = 1.88, *p* = 0.18
“Genotype” × “Treatment”	F(1,27) = 4.0, *p* = 0.055	F(1,27) = 2.16, *p* = 0.08

## Data Availability

The data presented in this study are available in this article.

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
