# Peer review of "In Vitro and In Vivo Chaperone Effect of (R)-2-amino-6-(1R, 2S)-1,2-dihydroxypropyl)-5,6,7,8-tetrahydropterin-4(3H)-one on the C1473G Mutant Tryptophan Hydroxylase 2"

_biomolecules, 2023, doi:10.3390/biom13101458_

Round 1

Reviewer 1 Report

The manuscript entitled “In vitro and in vivo chaperone effect of (R )-2-amino-6-(1R, 2S)-1,2-hihydroxypropyl)-5,6,7,8-tetrahydropterin-4(3H)-one on the C1473G mutant tryptophan hydroxylase 2” submitted by Kulikov reported some interested results.  In vitro, (R )-2-amino-6-(1R, 2S)-1,2-hihydroxypropyl)-5,6,7,8-tetrahydropterin-4(3H) (BH4) increased thermal stability of the mutant and wild type TPH2 molecules. But, intraventricular administration of BH4 did not alter the mutant TPH2 activity in the brain of Balb/c mice. This result indicates that although BH4 shows chaperone effect in vitro, it is unable to increase the activity of mutant TPH2 in vivo. Although the findings in this report does not provide positive use of BH4 to increase TPH2 activity, the results pointed out the difference effect between in vitro and in viva studies, may suggest more studies are needed to understand the underneath mechanism.

No major language errors are found.

Author Response

The manuscript entitled “In vitro and in vivo chaperone effect of (R )-2-amino-6-(1R, 2S)-1,2-hihydroxypropyl)-5,6,7,8-tetrahydropterin-4(3H)-one on the C1473G mutant tryptophan hydroxylase 2” submitted by Kulikov reported some interested results.  In vitro, (R )-2-amino-6-(1R, 2S)-1,2-hihydroxypropyl)-5,6,7,8-tetrahydropterin-4(3H) (BH4) increased thermal stability of the mutant and wild type TPH2 molecules. But, intraventricular administration of BH4 did not alter the mutant TPH2 activity in the brain of Balb/c mice. This result indicates that although BH4 shows chaperone effect in vitro, it is unable to increase the activity of mutant TPH2 in vivo. Although the findings in this report does not provide positive use of BH4 to increase TPH2 activity, the results pointed out the difference effect between in vitro and in viva studies, may suggest more studies are needed to understand the underneath mechanism.

Answer: This phrase was added to the Discussion section (lines 397-404)

Comments on the Quality of English Language

No major language errors are found.

Answer: English was improved by a professional proofreader.

Thank you very much for your valuuable comments

Reviewer 2 Report

The mechanisms and therapy of hereditary psychopathologies are an important problem inmodern neurogenetics and molecular neurophysiology. The cause of a number of severepsychopathologies are non-synonymous amino acid substitutions in the molecules of metabolicenzymes, transporters and receptors of neurotransmitters and mediator receptors which alter the3D structures of these molecules. Application of pharmacological chaperones - short moleculescorrecting the 3D structure of target mutant macromolecules - are promising although poorlystudied a promising direction in modern psychopharmacogenetics. Tetrahydrobiopterin (BH4)has been shown to be a promising pharmacological chaperone to correct hereditary alterations inphenylalanine and tyrosine hydroxylases. At the same time, application of BH4 for correction ofgenetic alterations in tryptophan hydroxylase 2 – the key and rate limiting enzyme of serotoninsynthesis in the mammalian brain – is still obscure.In the paper of Komleva et al. investigates an ability of BH4 to correct genetically definedalterations in the TPH2 activity in vitro and in vivo. The authors used the C1473G mutation inthe mouse Tph2 gene that results in twofold reduce in the TPH2 activity as a model to test BH4for its chaperone activity. Although this mutation causes only moderate alteration in the enzymeactivity it is the most available model since it is found in a common mouse strain such as Balb/c.In the experiment in vitro the authors showed that BH4 increased the TPH2 thermal stability i.e.it demonstrated a chaperone-like activity. In the experiments in vivo they showed that BH4administered ip did not penetrate brain and did not correct the reduced TPH2 activity and hindlimb dystonia caused by the 1473G mutation in Balb/c mice. Moreover, being injected directlyinto the brain ventricles, this drug is very quickly utilized and does not have time to show itschaperone effect.Since BH4 is widely advertised as a treatment for phenylketonuria, the negative results obtainedby the authors show the ineffectiveness of this drug for the treatment of mental disorders due toits low ability to penetrate into the brain.Comments1. Introduction. How many functional mutations in the Tph2 gene are known?2. Methods. What are the reasons for choosing the hippocampus and midbrain? Why CD-1 miceare used to study BH4 penetration the brain rather than C57BL/6 or Balb/c mice?3. Discussion. Exogenous BH4 does not enter the brain and is rapidly degraded. Are there waysto increase the synthesis of endogenous BH4?

Author Response

Comments

  1. Introduction. How many functional mutations in the Tph2 gene are known?

Answer: Information about known functional mutations in rthe Tph2 gene was include in the Introduction section (lines 46-68).

  1. Methods. What are the reasons for choosing the hippocampus and midbrain? Why CD-1 mice

are used to study BH4 penetration the brain rather than C57BL/6 or Balb/c mice?

Answer: The midbrain and hippocampus were chosen because these structure contain the 5-HT neurons’ bodies and endings respectively (lines 149-152). CD-1 mice were chosen as a “neutral” mouse model to test the BH4 penetrance to the brain (lines 89-91).

  1. Discussion. Exogenous BH4 does not enter the brain and is rapidly degraded. Are there ways

to increase the synthesis of endogenous BH4?

Answer: The level of endogenous BH4 in the brain can be increased by administration of BH4 metabolic precursors (lines 407-408).

Thank you very much for your comments.

Reviewer 3 Report

Tetrahydrobiopterin (BH4) acts as a co-factor of several aromatic amino acid hydroxylases, including the enzyme Tph2, the limiting enzyme on serotonin synthesis. BH4 has been used to treat several conditions, including ASD, ADHD, and depression, suggested to aid Tph2 function and 5-HT production with some studies suggesting this could be the case. In this new report, Arefieva et al. test the effect of BH4 administration in the in vitro and in vivo Tph2 function. The authors provide interesting results suggesting a limited brain availability of BH4, prompting interesting questions regarding the effectiveness in short, one-week, administration. Nonetheless, there are some considerations that I believe need to be addressed.  

It took me a while to find important information in the text. For instance, why use the Balb/c mice? This should be in the introduction of the manuscript instead of the discussion.

Also, on this point, I believe that choosing the C57BL/6 mice as a control for the Balb/c mice is not ideal. There are many genetic differences between the two strains and therefore comparisons between both mice strains, particularly 5-HT levels, could not be attributed only to Tph2 mutations in in vivo experiments. However, I do believe the work has the merit of showing any effect of BH4 administration on independent lines. I suggest the authors tone down the aim, discussion, and conclusions alluding to the effect of BH4 on correcting Tph2 mutations. 

Fig 3 missing panel letter

The way the authors determined the thermal stability is interesting. Although it is a proxy, considering it is not pure protein, it is still very informative. I would highly suggest the authors include references to literature using this protocol (e.g. Biochem Mol Biol Educ. 2018 Jul;46(4):398-402; Biotechnol Bioeng. 1993 Nov 20;42(10):1245-51) and include the complete unfolding profiles Tm vs. Activity graphs.  

Some parts of the manuscript lack coherence and were difficult to read. 

Author Response

Tetrahydrobiopterin (BH4) acts as a co-factor of several aromatic amino acid hydroxylases, including the enzyme Tph2, the limiting enzyme on serotonin synthesis. BH4 has been used to treat several conditions, including ASD, ADHD, and depression, suggested to aid Tph2 function and 5-HT production with some studies suggesting this could be the case. In this new report, Arefieva et al. test the effect of BH4 administration in the in vitro and in vivo Tph2 function. The authors provide interesting results suggesting a limited brain availability of BH4, prompting interesting questions regarding the effectiveness in short, one-week, administration. Nonetheless, there are some considerations that I believe need to be addressed. 

  1. It took me a while to find important information in the text. For instance, why use the Balb/c mice? This should be in the introduction of the manuscript instead of the discussion.

Answer: The information concerning the C and G distribution among laboratory mouse lines was added to the Introduction section (lines 66-68).

  1. Also, on this point, I believe that choosing the C57BL/6 mice as a control for the Balb/c mice is not ideal. There are many genetic differences between the two strains and therefore comparisons between both mice strains, particularly 5-HT levels, could not be attributed only to Tph2 mutations in in vivo experiments. However, I do believe the work has the merit of showing any effect of BH4 administration on independent lines.

Answer: You are right, C57BL/6 is not a correct control for Balb/c mice. We corrected this mistake: in the present study Balb/c and C57BL/6 mice are used only as models of mutant and wild type TPH2 (lines 89-91)

  1. I suggest the authors tone down the aim, discussion, and conclusions alluding to the effect of BH4 on correcting Tph2 mutations.

Answer: You are right. The aims and discussion sections were corrected and toned down. At the same time, the conclusion section includes only experimental facts without any general reasoning.

  1. Fig 3 missing panel letter

Answer: The figure 3 was corrected.

  1. The way the authors determined the thermal stability is interesting. Although it is a proxy, considering it is not pure protein, it is still very informative. I would highly suggest the authors include references to literature using this protocol (e.g. Biochem Mol Biol Educ. 2018 Jul;46(4):398-402; Biotechnol Bioeng. 1993 Nov 20;42(10):1245-51) and include the complete unfolding profiles Tm vs. Activity graphs.

Answer: The Tm vs Activity graphs (A and B) were added to the figure 1. In the present study we used the modified protocol applied for TPH, TH and PAH studies [11-13]. This protocol differs from the protocols of Hei, Clark (1993) and Saqib and Siddiqui (2018). The first protocol used only the final part of the Tm vs Activity curve and it is a simplified variant of the second protocol. At the same time, the first protocol is correct and more suitable for the T50 calculation.

Comments on the Quality of English Language

Some parts of the manuscript lack coherence and were difficult to read.

Answer: English was improved by a professional proofreader.

Thank you much for your valuable comments.

Reviewer 4 Report

The manuscript is well-written and its content is clearly exposed. I do, however, have a single observation to make:

In their work, the authors exclusively reference Thony et al. 2008 in contrast to their own findings. I recommend that the author broaden their discussion by reading the comprehensive review by Fanet et al. 2021 (Tetrahydrobioterin (BH4) Pathway: From Metabolism to Neuropsychiatry, Current Neuropharmacology, Volume 19(5), published on April 29, 2021). Incorporating insights from this review into their discussion would enable them to more effectively address the contrasting results. Notably, Table 1 of this review lists 18 studies investigating the effect of BH4 administration on cerebral monoamines.

Author Response

The manuscript is well-written and its content is clearly exposed. I do, however, have a single observation to make:

In their work, the authors exclusively reference Thony et al. 2008 in contrast to their own findings. I recommend that the author broaden their discussion by reading the comprehensive review by Fanet et al. 2021 (Tetrahydrobioterin (BH4) Pathway: From Metabolism to Neuropsychiatry, Current Neuropharmacology, Volume 19(5), published on April 29, 2021). Incorporating insights from this review into their discussion would enable them to more effectively address the contrasting results. Notably, Table 1 of this review lists 18 studies investigating the effect of BH4 administration on cerebral monoamines.

Answer: Thank you very much for this very valuable reference [40]. This reference as well as another reference of Fanet et al [41] together with Thony [14] forced us to reconsider our conclusion concerning efficacy of BH4 therapy and we toned down our general reasoning about this therapy (lines 397-404).

We thank you very much for your valuable comments.

Round 2

Reviewer 3 Report

The authors have answered all of my concerns.